# High-Density Patterned Array Bonding through Void-Free Divinyl Siloxane Bis-Benzocyclobutene Bonding Process

**DOI:** 10.3390/polym13213633

**Published:** 2021-10-21

**Authors:** Nam Woon Kim, Hyeonjeong Choe, Muhammad Ali Shah, Duck-Gyu Lee, Shin Hur

**Affiliations:** 1Department of Nature-Inspired System and Application, Korea Institute of Machinery and Materials, Daejeon 34103, Korea; a79428209@nate.com (N.W.K.); ali@kimm.re.kr (M.A.S.); 2Department of Drug Discovery, Korea Research Institute of Chemical Technology, Daejeon 34114, Korea; lastlove2ya@naver.com; 3Department of Nano-Mechatronics, University of Science and Technology, Daejeon 34113, Korea

**Keywords:** DVS-BCB bonding mechanism, DVS-BCB bonding, adhesive, void-free DVS-BCB bonding, pressure condition

## Abstract

Divinylsiloxane-bis-benzocyclobutene (DVS-BCB) has attracted significant attention as an intermediate bonding material, owing to its excellent properties. However, its applications are limited, due to damage to peripheral devices at high curing temperatures and unoptimized compressive pressure. Therefore, it is necessary to explore the compressive pressure condition for DVS-BCB bonding. This study demonstrates an optimization process for void-free DVS-BCB bonding. The process for obtaining void-free DVS-BCB bonding is a vacuum condition of 0.03 Torr, compressive pressure of 0.6 N/mm^2^, and curing temperature of 250 °C for 1 h. Herein, we define two factors affecting the DVS-BCB bonding quality through the DVS-BCB bonding mechanism. For strong DVS-BCB bonding, void-free and high-density chemical bonds are required. Therefore, we observed the DVS-BCB bonding under various compressive pressure conditions at a relatively low temperature (250 °C). The presence of voids and high-density crosslinking density was examined through near-infrared confocal laser microscopy and Fourier-transform infrared microscopy. We also evaluated the adhesion of the DVS-BCB bonding, using a universal testing machine. The results suggest that the good adhesion with no voids and high crosslinking density was obtained at the compressive pressure condition of 0.6 N/mm^2^. We believe that the proposed process will be of great significance for applications in semiconductor and device packaging technologies.

## 1. Introduction

Owing to the development of the semiconductor industry, bonding wafers with complex structures of semiconductors and micro-electromechanical systems (MEMS) devices without interconnection delay, low adhesive strength, crosstalk, and power dissipation have attracted significant attention [1,2]. In particular, with increasingly precise device structures, it is imperative to maintain good junctions, even in narrow spans, such as between patterned arrays. Generally, processes such as metal eutectic bonding, metal thermocompression bonding, silicon fusion bonding, anodic bonding, and adhesive bonding are used to bond wafers [3]. Adhesive bonding is one of the main processes with the advantages of high yield, low cost, high bonding strength, relatively low bonding temperature, wide applicability to various wafer materials, no requirement for electric voltage, and complementary metal–oxide–semiconductor and MEMS compatibility [3,4]. This adhesive wafer bonding uses various polymers, such as polyimide [5], photoresist [6], epoxy [7], and divinyl siloxane bis-benzocyclobutene (DVS-BCB) [8,9] as an intermediate bonding material. In particular, the commercial dielectric polymer, i.e., DVS-BCB, has attracted significant attention, owing to its excellent properties, such as low dielectric constant (~2.65 for the frequency range of 10 Hz to 1 MHz), low moisture absorption, low cure temperature, high degree of planarization, low level of ionic contaminants, high optical clarity, good thermal stability, excellent chemical resistance, and good compatibility with various metallization systems [10,11,12,13,14]. Therefore, DVS-BCB is used in bonding and packaging processes for three-dimensional (3D) integrated circuit [4,15], optical [16,17], and MEMS devices [17,18,19].

To obtain high-quality wafer bonding, it is crucial to develop a uniform and void-free bonding process. Common polymer adhesives deform to a liquid-like state at high temperatures and high pressures, and thus, they flow and accommodate on the wafer surface to achieve wafer bonding [20]. The deformability and flow capabilities of several polymers, including DVS-BCB, offer high bond yields and high bonding strength for a variety of wafer topographies [20]. However, if the air trapped in the polymer is not completely removed during the wafer bonding process, voids that act as defects are formed. Voids resulting from non-bonding at the wafer bonding interface can affect high local stress, low bonding strength, poor sealing performance, low yield, and poor reliability [21,22,23]. Void formation in wafer bonding may be caused by surface inactivity [24], surface contaminants [25], bonding layer interdiffusion [22] and inappropriate bonding parameters [26]. Special processes, such as O_2_ plasma activation [24], wet cleaning [25], anti-interdiffusion [22], and bonding process optimization [26], are required to suppress factors that result in void formation. In particular, the void formation mechanism that weakens the bonding strength in adhesive wafer bonding is mostly affected by the curing temperature and compressive pressure process [7]. Lin et al. reported that the bonding temperature should be controlled for high DVS-BCB bonding adhesion, and the optimum temperature for DVS-BCB for oxide bonding at a compression strength of 50 N was 300 °C [15]. Xu et al. demonstrated a spot-pressing bonding technique with a strength of 0.03 or 0.05 MPa to completely remove the voids generated during the bonding process [27]. However, void-free DVS-BCB bonding has limitations, such as damage to peripheral devices, due to high bonding temperatures at unoptimized compressive pressure. Therefore, it is necessary to explore the compressive pressure condition for void-free DVS-BCB bonding. Moreover, it is important to study the physicochemical behavior of DVS-BCB bonding.

In this study, we present a void-free optimal compressive pressure process, using a commercial DVS-BCB to bond with silicon wafers with a high-density pattern array. DVS-BCB bonding was performed in a vacuum environment of 0.03 Torr at a temperature of 250 °C and different compressive pressures of 0.4, 0.5, and 0.6 N/mm^2^. The shape of the DVS-BCB bonding pattern and the presence or absence of voids were studied through a non-destructive analysis, conducted using a near-infrared confocal laser microscope. In addition, Fourier-transform infrared (FT-IR) microscopy was performed to investigate the chemical bonding relationship between the silicon wafer, AP 3000, and DVS-BCB, according to the compressive pressure. Consequently, it was confirmed that the DVS-BCB bonding with no voids and good chemical bonding between the polymers had the strongest adhesion. This proposed DVS-BCB bonding process is believed to contribute to the packaging and applications of 3D integrated circuit, optical, and MEMS devices without bonding defects.

## 2. Materials and Methods

### 2.1. DVS-BCB Pattern Process on a Silicon Wafer

A 6 inch p-type single-side polished silicon wafer (resistivity = 1–10 ohm/cm) with a thickness of 675 ± 25 μm was prepared for use in the DVS-BCB pattern. After dropping an adhesion promoter (AP 3000, Dow Chemical, Midland, MI, USA) on the prepared silicon wafer, spin coating was performed at 500 rpm for 5 s and then at 3000 rpm for 30 s. Subsequently, heat treatment was performed at 100 °C for 1 min, using a hot plate. After dropping the DVS-BCB solution (Cyclotene 4022-25, Dow Chemical, MI, USA) onto a silicon substrate coated with AP 3000, spin coating was performed at 500 rpm for 5 s and then at 3000 rpm for 30 s. When the DVS-BCB solution coating was completed, a soft bake was performed at 100 °C for 1 min, using a hot plate. We fabricated a glass mask coated with a Cr pattern to fabricate a device-like 3D DVS-BCB structure. The DVS-BCB-coated silicon wafer was exposed to a broadband ultraviolet (UV) light, using a fabricated high-quality glass mask and a mask aligner (MA-6 with UV 400 optics, Karl Suss, Garching, Germany). The mask aligner had an exposure time of 100 ms using an i-line (365 nm) filter at a UV light intensity of 17 mW/cm^2^. The fabricated wafer samples were post-exposure baked at 100 °C for 1 min, and then developed at 30 °C for 1 min, using a developer (DS-3000, Dow Chemical, MI, USA).

### 2.2. DVS-BCB Bonding Process

We prepared two silicon wafers coated with a 3D DVS-BCB structure and aligned them with an aligner (HAS-200, Hutem, Seoul, Korea) using the alignment key demonstrated in Appendix A. As shown in Appendix A, the DVS-BCB bonding process was performed, using the optimal curing temperature process provided by Dow Chemical [28]. To determine the optimized void-free DVS-BCB bonding process, the curing and compression processes were performed using a compression wafer bonding system (HBS-400, Hutem, Seoul, Korea). For DVS-BCB bonding, compressing pressures of 0.4, 0.5, and 0.6 N/mm^2^ were applied for 1 h when the curing temperature reached 250 °C in a vacuum condition of 0.03 Torr. Thereafter, the bonding system was gradually cooled to 25 °C. A schematic of the DVS-BCB pattern and bonding process is presented in Figure 1.

### 2.3. DVS-BCB Bonding Characterization

The shape of the coated DVS-BCB pattern layer was observed, using an optical microscope (BX-41, Olympus, Tokyo, Japan). Raman spectroscopy (Raman, DXR, Thermo Fisher Scientific, Waltham, MA, USA) was performed to confirm the removal of the DVS-BCB material during the DVS-BCB pattern formation process. To determine the voids and pattern deformation of the DVS-BCB bonding layer, a non-destructive analysis was performed, using a near-infrared confocal laser microscope (LEXT OLS-3000, Olympus, Tokyo, Japan). Scanning electron microscopy (SEM, Nova Nano SEM200, FEI, Hillsborough, OR, USA) was performed to observe changes in the thickness of the DVS-BCB bonding layer with respect to the compressive pressure. FT-IR microscopy (Nicolet iN10 Infrared Microscope, Thermo Fisher Scientific, MA, USA) was performed to observe the chemical bonding properties of the DVS-BCB bonding layer produced, according to the compressive pressure.

### 2.4. DVS-BCB Bonding Adhesion Analysis

The adhesive strength characteristics of the DVS-BCB bonding produced according to the compressive pressure were confirmed, using a universal testing machine (Model 5982, Instron, Norwood, MA, USA). To connect the DVS-BCB bonding sample to the universal testing machine, a two-step jig was prepared and analyzed. First, an aluminum jig with dimensions of 12.5 mm × 32 mm × 14.5 mm (w × l × h) was manufactured, and the DVS-BCB bonding sample was attached using AXIA EP-04 epoxy (AXIA EP-04, Alteco, Gyeonggi-do, Korea) (Appendix A). For strong adhesion, the epoxy was dried at room temperature for 1 day. Thereafter, the strongly bonded DVS-BCB bonding sample and the aluminum jig were connected to a universal testing machine by making a stainless steel upper and lower connection jig (Appendix A). The adhesion test for DVS-BCB bonding was conducted at a speed of 1 mm/min, and only the results of detaching the DVS-BCB bonding layer were used.

## 3. Results and Discussion

### 3.1. DVS-BCB Bonding Mechanism

For DVS-BCB bonding with strong adhesion, it is necessary to understand the chemical reaction and chemical bonding of the materials used and to minimize the defects that may occur. In Figure 2, we identified the chemical structures of the materials used for DVS-BCB bonding and show the chemical reactions that occur during the bonding process. For high adhesion between DVS-BCB and silicon wafer, adhesion promoter (AP 3000) was coated on a silicon wafer, and then the DVS-BCB solution was additionally coated. AP 3000 is vinyltriacetoxysilane [CH_2_CHSi(OCOCH_3_)_3_] (chemical structure shown in Figure 2a), which contains one vinyl (CH_2_CH) group and three acetoxy (OCOCH_3_) groups [29]. The acetoxy groups undergo a hydrolysis reaction with water to form hydroxyl groups (Si–OH) in AP 3000 (Figure 2a), followed by a condensation reaction with Si–OH in AP 3000 or Si–OH on the silicon wafer surface to create strong siloxane bonds (Si–O–Si) (Figure 2b,c) [29]. In general, the presence of bridging bonds increases the critical adhesion energy. In the absence of AP 3000, DVS-BCB and the silicon wafer surface mainly form chain entanglement physical bonding. Therefore, for high-quality DVS-BCB bonding, it is necessary to form a strong DVS-BCB bonding layer through many bridging bonds, using AP 3000. However, when AP 3000 was used, water in AP 3000 and that generated by the condensation reaction of AP 3000 could be vaporized to form voids. Because these voids act as defects in DVS-BCB bonding, it is necessary to remove them through an optimized compressive process [20,24,30]. In DVS-BCB, as shown in Figure 2d, cyclobutene with a four-membered ring structure undergoes a rotational ring-opening reaction at approximately 180 °C or higher to transform to o-xylylene [31,32,33,34]. Because of these properties, the resulting o-xylylene has a very high reactivity and forms a bond with the vinyl groups of AP 3000, as shown in Figure 2e. In addition, o-xylylene easily undergoes Diels–Alder reactions with the vinyl structure of the DVS part, resulting in bonding between DVS-BCB units or DVS-BCB layers (Figure 2f). As shown in Figure 2f, because the number of o-xylylene rings and vinyl functional groups is stoichiometric in the DVS-BCB molecule, most of the unsaturated bonds in the DVS-BCB units are consumed during thermal crosslinking. As such, the DVS-BCB curing process does not require a catalyst, and it is a thermal process that does not generate by-products [32].

We defined two factors that affect the strength of DVS-BCB bonding through chemical reactions that occur during the bonding process of DVS-BCB. First, all voids created by the vaporization of H_2_O in the AP 3000 layer must be removed from the DVS-BCB layer. Second, it is necessary to create a high-density polymer with ideal chemical bonding of each material at an optimized curing temperature and compression process. We proposed an optimal void-free DVS-BCB bonding process by investigating the DVS-BCB bonding products manufactured, according to the compression conditions based on these two factors. Figure 3 shows a simplified schematic illustration of the reaction occurring at the interface of each coated material.

### 3.2. Characteristics of 3D-Patterned DVS-BCB Bonding Layers

We measured optical microscopy images, as shown in Figure 4, to observe the patterned DVS-BCB layer exposed to the UV light passing through the glass mask. Figure 4a shows a schematic of manufacturing a glass mask, and 177 pattern arrays with a pitch of 107.2 μm are arranged in one line to obtain a total of 708 pattern arrays. In addition, a glass mask was prepared by arranging an alignment key for alignment at both ends of the center. Optical microscopic images of the 3D DVS-BCB pattern fabricated using the prepared glass mask are shown in Figure 4b,c. As shown in Figure 4b, the manufactured DVS-BCB pattern was produced in a uniformly well-aligned pattern without deformation. In addition, as shown in Figure 4c, the DVS-BCB pattern and the developed interface are identical and have a clean shape. The fabricated DVS-BCB layer was subjected to Raman spectroscopy to determine whether all the DVS-BCB layers inside the pattern were removed through the development process. From Appendix A, it can be confirmed that all the DVS-BCBs were removed by the development process [35]. This confirms that the 3D DVS-BCB layer prepared for DVS-BCB bonding was fabricated with high quality without defects or shape collapse.

Figure 5 shows the analysis results of the DVS-BCB bonding samples manufactured with various compressive pressures at a curing temperature of 250 °C through precise alignment using two patterned DVS-BCB silicon wafers. In the DVS-BCB curing process, the curing temperature has a very important effect. The Dow Chemical Company announced that DVS-BCB material cures more than 98% under curing conditions of 1 h at 250 °C. In addition, it is suggested that a long curing time of 60 h or more is required at a curing temperature of 200 °C [28]. DVS-BCB materials cure rapidly at high curing temperatures such as 300 °C, but the high temperatures can damage other materials and structures. Therefore, we adjusted the compression pressure by fixing the condition for 1 h at 250 °C, which is the optimized curing condition provided by Dow Chemical [28]. We investigated the morphological deformations and defects generated in the DVS-BCB bonding process using near-infrared confocal laser microscopy. Figure 5a–c shows a single-pattern shape as a result of DVS-BCB bonding produced under compressive pressure of 0.4, 0.5, and 0.6 N/mm^2^, confirming that the two DVS-BCB silicon wafers are well aligned and bonded. In addition, it was confirmed that the voids generated by the water of the AP 3000 layer did not fall out of the DVS-BCB layer and still existed since the compressive pressure was lower than 0.6 N/mm^2^. In particular, when the compressive pressure was 0.4 N/mm^2^, it was confirmed that a large number of voids exist between the patterns, which is a path through which it is difficult for the pores to escape. When the compressive pressure was 0.5 N/mm^2^, several voids escaped outside the DVS-BCB layer, but it was confirmed that some voids were still present at some points. However, it was confirmed that void-free DVS-BCB bonding can be obtained at a compressive pressure of 0.6 N/mm^2^. Because DVS-BCB has high fluidity at high temperatures, voids are released to the outside of the layer through ideal compression under vacuum conditions. If these voids remain, they affect the bonding strength of the DVS-BCB bonding and eventually lead to cleavages. From the near-infrared confocal laser microscopy images obtained in Figure 5a–c, it was confirmed that the optimal compressive pressure to remove all voids between the high-density patterned arrays was 0.6 N/mm^2^. Figure 5d–f shows a DVS-BCB layer bonded between two silicon wafers. SEM analysis of the sample cross-section was performed to investigate the thickness deformation according to various compressive pressures. From the SEM image, it was confirmed that the DVS-BCB layer had a thickness of ~2.2 μm and was uniform without a significant difference at all compressive pressures. From this, it was confirmed that the pattern shape and the thickness of the DVS-BCB layer did not change, even at high pressures, such as at a compressive pressure condition of 0.6 N/mm^2^ without voids. Therefore, we found that the optimal void-free DVS-BCB bonding condition is 0.6 N/mm^2^, wherein voids do not exist and the pattern shape and DVS-BCB layer thickness do not deform.

We conducted an FT-IR analysis to determine the effects of various compressive pressure conditions on the chemical bonding behavior in the DVS-BCB bonding process. For accurate comparison of the samples, a portion with the same thickness was found in the exposed DVS-BCB layer through precise cutting, and it was measured, using an FT-IR microscope (spot size 80 μm × 80 μm). As shown in Figure 6, DVS-BCB of the prepared specimens was investigated at ~1600 cm^-1^, which is the core fingerprint region of the molecule. Owing to the relatively large DVS-BCB molecule, there are numerous partially overlapping peaks in the fingerprint region [15]. As shown in Figure 6, four distinct and strong absorption peaks were observed for the DVS-BCB layer. The prominent features in the FT-IR spectra are Si–CH_3_ stretch (≈1255 cm^−1^), Si–O–Si stretch (≈1040 cm^−1^), Si–O bending (≈830 cm^−1^), and Si–CH_3_ bending (≈780 cm^−1^), which is consistent with the results of typical DVS-BCB bonding [36,37]. With these features, we can observe the degree of Si–O–Si bonding and Si–CH_3_ bonding. The analysis confirmed that the absorption peak increased as the compressive pressure increased from 0.4 N/mm^2^ to 0.6 N/mm^2^. The increase in the absorption peak intensity indicates an increase in the Si–O–Si and Si–CH_3_ bond densities based on the same area, thereby suggesting that more uniform and more bridged bonds were generated. These results are directly related to the adhesive strength of DVS-BCB bonding, and it is expected that the strongest and most stable bonding is achieved at 0.6 N/mm^2^. We report the importance of the degree of chemical bonding in DVS-BCB bonding, in addition to the presence or absence of voids, as a factor that significantly affects the adhesion strength of DVS-BCB bonding.

### 3.3. Adhesive Strength Characteristics of DVS-BCB Bonding

We confirmed the presence of voids and the behavior of chemical bonds in the DVS-BCB bonding fabricated according to the compressive pressure through near-infrared confocal laser microscopy and FT-IR analysis. We conducted a tensile test using a universal testing machine to investigate the effect of these properties on the adhesive strength of the DVS-BCB bonding. In the tensile test, an aluminum jig with dimension of 12.5 mm × 32 mm × 14.5 mm (w × l × h) and the prepared sample were strongly attached, using AXIA EP-04 (tensile shear strength of 2962 N/cm^2^) epoxy. We used epoxy with high adhesion strength to cut the DVS-BCB layer and confirmed that the DVS-BCB bonding layer was detached, as shown in Appendix A. Figure 7 shows the results of the adhesion test of the prepared sample, and it was confirmed that the adhesion strengths were 8.82, 11.13, and 11.50 N/cm^2^ at compressive pressures of 0.4, 0.5, and 0.6 N/mm^2^, respectively. These results are consistent with the results of Figure 5 and Figure 6, wherein the highest adhesive strength is expected at the compressive pressure of 0.6 N/mm^2^. The adhesive strength at 0.4 N/mm^2^ is lower than those at 0.5 and 0.6 N/mm^2^, owing to the presence of a large number of voids and low chemical bonding. From the results of 0.5 and 0.6 N/mm^2^, it can be seen that the increase in the adhesion strength line shape with respect to the displacement is similar. Furthermore, it was confirmed that the compressive pressure of 0.6 N/mm^2^ resulted in greater adhesion strength by 0.37 N/cm^2^ than the compressive pressure of 0.5 N/mm^2^; this phenomenon can be attributed to the difference in the number of chemical bonds rather than the presence of voids generated during the DVS-BCB bonding process. Based on these results, we confirmed that the morphological, chemical, and physical analysis results of the DVS-BCB bonding are in good agreement. In the DVS-BCB bonding process, the high compressive pressure induces high stress on the high-density pattern array. These high stresses can destroy or crack the high-density pattern array structure. Therefore, we propose that the optimized compressive pressure condition for void-free DVS-BCB bonding is 0.6 N/mm^2^. We carried out near-infrared confocal laser microscopy to examine the presence of voids even in DVS-BCB bonding with different patterns under the proposed compressive pressure condition of 0.6 N/mm^2^. It was confirmed that there was no void in the super high-density patterns and linear patterns, as shown in Appendix A, respectively. This indicates that our proposed DVS-BCB bonding process has excellent reproducibility and good bonding in various patterns.

## 4. Conclusions

In this study, the shape and physicochemical properties of DVS-BCB bonding according to the compressive pressure were investigated to determine the optimal void-free DVS-BCB bonding process conditions. The optimal process conditions result in good adhesion and void-free bonding, even in a narrow area such as a high-density pattern array. We propose the presence of voids and differences in crosslinking density as factors affecting DVS-BCB bonding through the bonding mechanism of DVS-BCB bonding. Through near-infrared confocal laser microscopy, FT-IR, and universal testing machine analysis, we found that the presence of voids and the crosslinking density of the materials have a significant effect on the bonding strength of the DVS-BCB bonds. We confirmed that the DVS-BCB bonding sample produced with a compression strength of 0.6 N/mm^2^ has the best adhesion, highest chemical bonding density, and no voids. Moreover, it is confirmed that the 3D structure of the DVS-BCB layer is not destroyed and the thickness of the DVS-BCB bonding layer is not deformed, thereby indicating no deformation of the DVS-BCB bonding layer by the compressive pressure. Based on our results, we found that the optimized void-free DVS-BCB bonding process is temperature curing at 250 °C for 1 h at a compressive pressure of 0.6 N/mm^2^ under vacuum conditions of 0.03 Torr. We expect that this proposed void-free DVS-BCB bonding process can be applied to the bonding and packaging processes of high-density MEMS device arrays, 3D integrated circuit devices, optical devices, and MEMS.

## Figures and Tables

**Figure 1 polymers-13-03633-f001:**
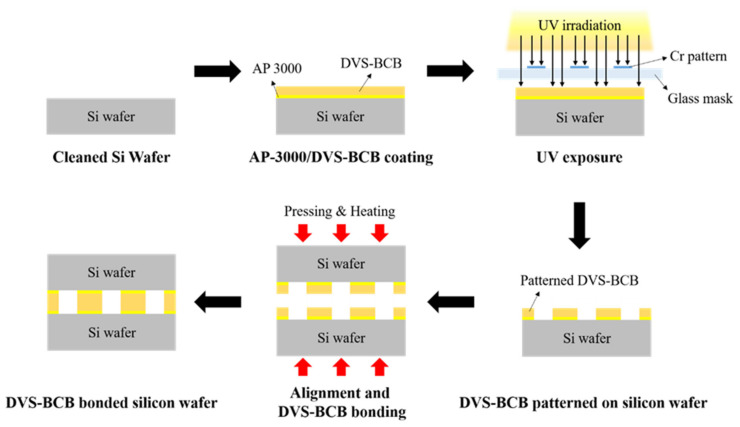
Schematic of the fabrication process for DVS-BCB bonding.

**Figure 2 polymers-13-03633-f002:**
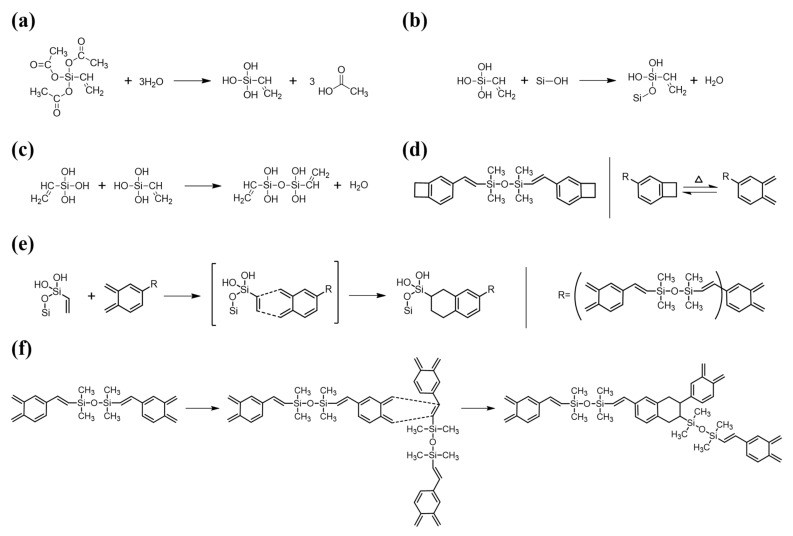
Reaction mechanism of AP 3000 and DVS-BCB during curing process. (**a**) Hydrolysis reaction of AP 3000, (**b**) Condensation reaction between hydrolyzed AP 3000 and silicon wafer surface, (**c**) Condensation reaction between hydrolyzed AP 3000s, (**d**) Chemical structure of DVS-BCB and ring-opening reaction of cyclobutene by heat, (**e**) Diels-Alder reactions between the vinyl group of hydroxylated AP 3000 (dienophile) and o-xylylene of DVS-BCB (diene), (**f**) Diels-Alder reactions between the vinyl group of DVS-BCB (dienophile) and o-xylylene of DVS-BCB (diene).

**Figure 3 polymers-13-03633-f003:**
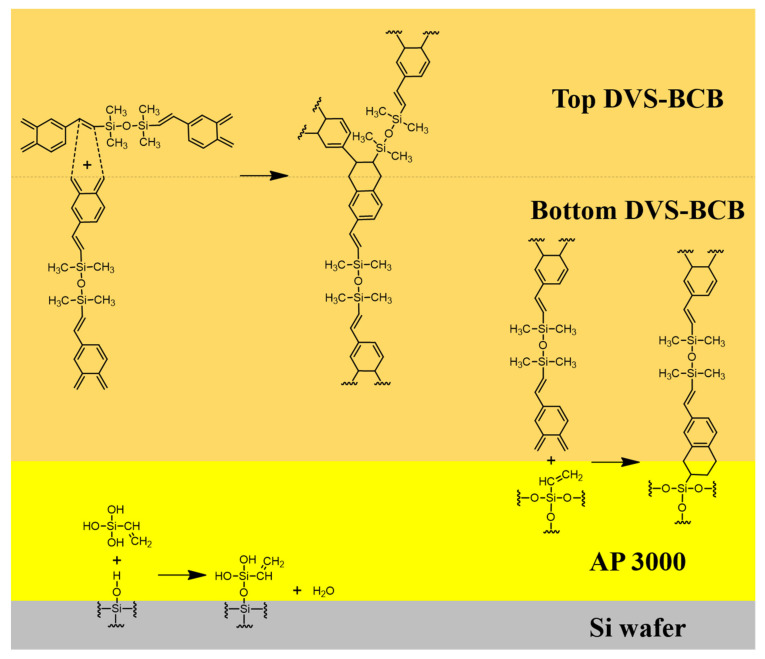
Reaction mechanism of the reaction occurring on the surface of each material during the curing process.

**Figure 4 polymers-13-03633-f004:**
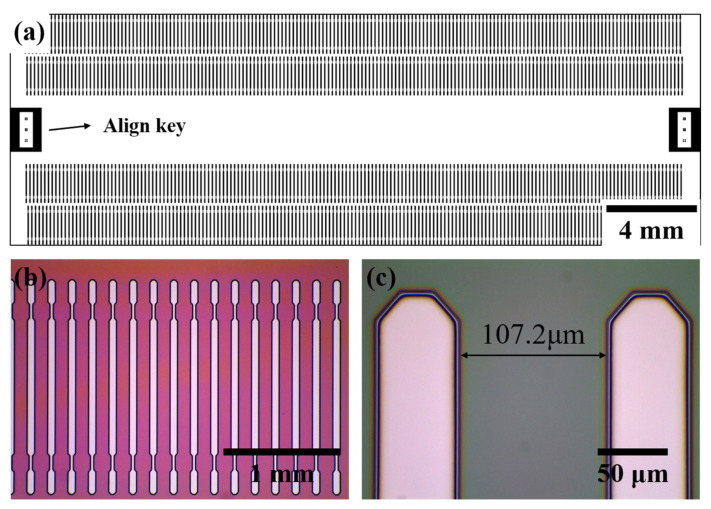
(**a**) Mask drawings for a 3D DVS-BCB structure pattern and (**b**,**c**) optical microscopic images of the fabricated 3D DVS-BCB structure.

**Figure 5 polymers-13-03633-f005:**
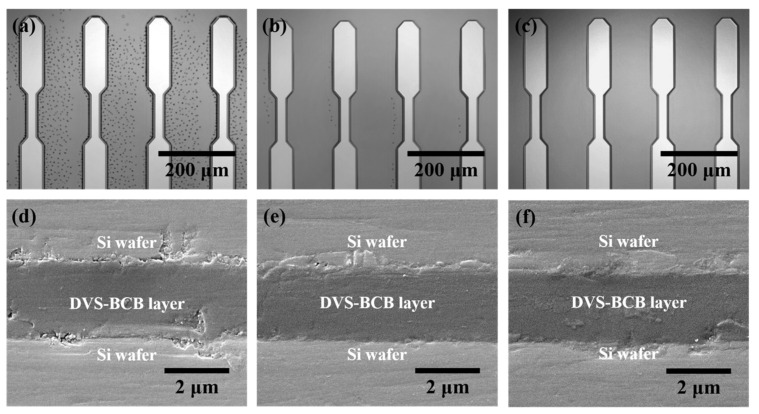
Analysis of the DVS-BCB bonding samples produced by different compressive pressures. (**a**–**c**) Near-infrared confocal laser microscopic images of the DVS-BCB bonding layer and (**d**–**f**) cross-sectional SEM images of bonded DVS-BCB bonding samples.

**Figure 6 polymers-13-03633-f006:**
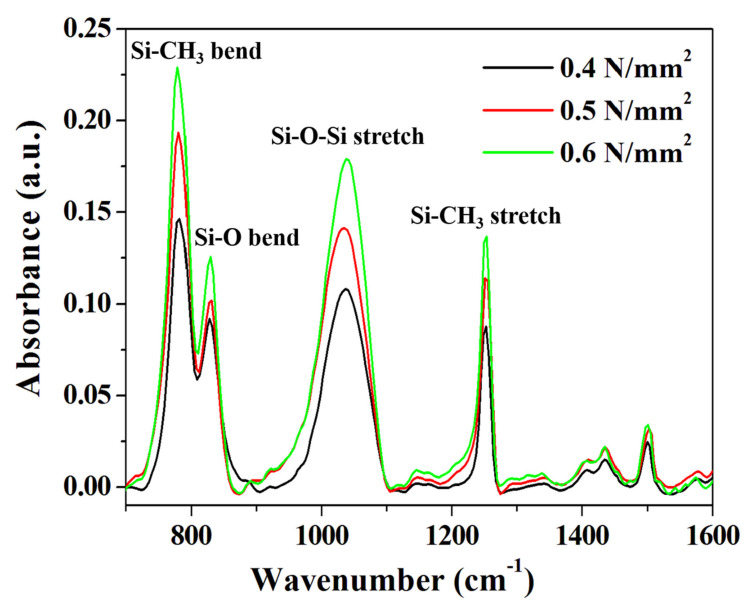
FT-IR absorbance spectra of the DVS-BCB bonding fabricated under various compressive pressure conditions.

**Figure 7 polymers-13-03633-f007:**
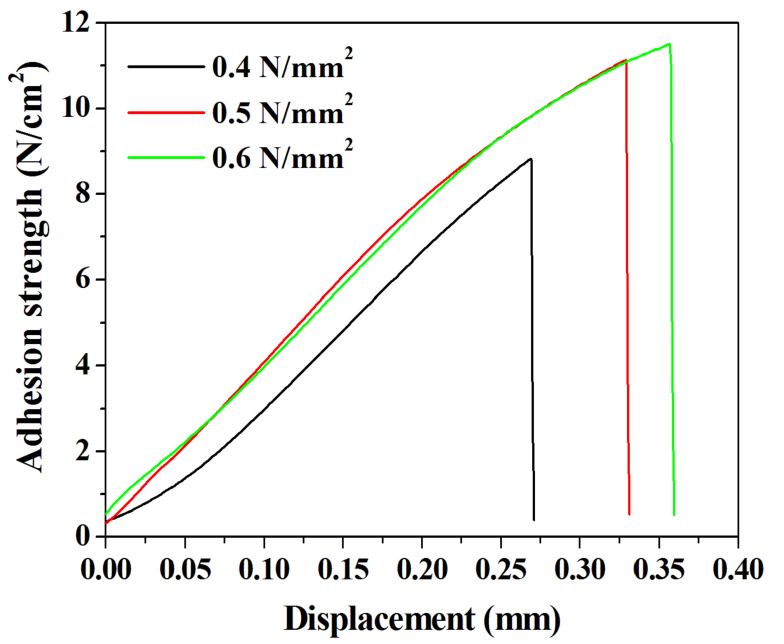
Adhesion test results of the DVS-BCB bonding samples prepared under various compressive pressure conditions.

## Data Availability

The data presented in this study are available in the article.

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
