# Peer review of "High-Density Patterned Array Bonding through Void-Free Divinyl Siloxane Bis-Benzocyclobutene Bonding Process"

_polymers, 2021, doi:10.3390/polym13213633_

Round 1
Reviewer 1 Report
- The manuscript investigated the optimization process for DVS-BCB bonding
- The authors concluded that the optimized DVS-BCB bonding process is temperature curing at 250 °C for 1 h at a compressive pressure of 0.6 N/mm2. The effect of the temperature on the performance of the DVS-BCB bonding was not included in the manuscript. It is necessary to explain why 250 °C was the optimized temperature.
3. Experimental results showed that the adhesion force is increasing as the compressive pressure is increasing from o.4 to 0.6 N/mm2 as shown in Figure 5. This does not mean that the 0.6 N/mm2 is the optimized compressive pressure. Unless, the authors can prove that adhesion force is decreasing as the compressive pressure is larger than 0.6 N/mm2.
Reviewer 2 Report
Line 23: high-density chemical bonds and Lines 312, 315: chemical bond density REMARK: “chemical bond density” is not used in polymer chemistry, please change to “crosslinking density”
REMARK please change everywhere in the text "adhesion force" to "adhesion strength"
Line 25: The results suggest that the compressive pressure condition of 25 0.6 N/mm2 with no voids and high chemical bond density had the highest adhesion CORRECT The results suggest that the highest adhesion with no voids and high crosslinking density was obtained at the compressive pressure condition of 0.6 N/mm2
Line 156: which consists of one vinyl… CORRECT: which contains of one vinyl…
Line 292: it was confirmed that the compressive pressure of 0.6 N/mm2 had a greater adhesion force by 0.37 N/cm2 than compressive pressure of 0.5 N/mm2 CORRECT it was confirmed that the compressive pressure of 0.6 N/mm2 resulted in greater adhesion strength by 0.37 N/cm2 than compressive pressure of 0.5 N/mm2
Author Response
Thank you for pointing out to us to improve. We are glad to improve our manuscript in response to your comments.
Line 23: high-density chemical bonds and Lines 312, 315: chemical bond density REMARK: “chemical bond density” is not used in polymer chemistry, please change to “crosslinking density”
- Thanks for pointing out the correct use of the nomenclature. Your comments have been very helpful in improving the quality of the manuscript. We changed "chemical bond density" in lines 23, 312 and 315 to "crosslinking density".
REMARK please change everywhere in the text "adhesion force" to "adhesion strength"
- Thanks for pointing out the correct use of the nomenclature. Your comments have been very helpful in improving the quality of the manuscript. We changed "adhesion force" on lines 285, 291, 293 and Figure 7 Y-axis to "adhesion strength".
Line 25: The results suggest that the compressive pressure condition of 25 0.6 N/mm2 with no voids and high chemical bond density had the highest adhesion CORRECT The results suggest that the highest adhesion with no voids and high crosslinking density was obtained at the compressive pressure condition of 0.6 N/mm2.
- Thank you for suggesting corrections to the sentences to convey the correct meaning. Your comments have been very helpful in improving the quality of the manuscript. We changed the sentence in line 25 to the sentence you suggested.
Line 156: which consists of one vinyl... CORRECT: which contains of one vinyl...
- Thanks for pointing out the correct use of the nomenclature. Your comments have been very helpful in improving the quality of the manuscript. We changed "consists" on line 156 to "contains".
Line 292: it was confirmed that the compressive pressure of 0.6 N/mm2 had a greater adhesion force by 0.37 N/cm2 than compressive pressure of 0.5 N/mm2 CORRECT it was confirmed that the compressive pressure of 0.6 N/mm2 resulted in greater adhesion strength by 0.37 N/cm2 than compressive pressure of 0.5 N/mm2.
- Thank you for suggesting corrections to the sentences to convey the correct meaning. Your comments have been very helpful in improving the quality of the manuscript. We changed the sentence in line 292 to the sentence you suggested.

Round 2
Reviewer 1 Report
The authors did not respond my comments appropriately. I recommend that the authors provide experimental results with compressive pressures of 0.7 and 0.8 N/mm2. A figure shown the adhesion strength vs. the compressive pressure from 0.4 to 0.8 N/mm2 can be used to determine whether 0.6 N/mm2 is the optimal pressure or not.
